# TRPV1 drugs alter core body temperature via central projections of primary afferent sensory neurons

Wendy Wing Sze Yue[1], Lin Yuan[1], Joao M Braz[2], Allan I Basbaum[2], David Julius[1]*

[1]Department of Physiology, University of California, San Francisco, United States; [2]Department of Anatomy, University of California, San Francisco, United States

**Abstract** TRPV1, a capsaicin- and heat-activated ion channel, is expressed by peripheral nociceptors and has been implicated in various inflammatory and neuropathic pain conditions. Although pharmacological modulation of TRPV1 has attracted therapeutic interest, many TRPV1 agonists and antagonists produce thermomodulatory side effects in animal models and human clinical trials, limiting their utility. These on-target effects may result from the perturbation of TRPV1 receptors on nociceptors, which transduce signals to central thermoregulatory circuits and release proinflammatory factors from their peripheral terminals, most notably the potent vasodilative neuropeptide, calcitonin gene-related peptide (CGRP). Alternatively, these body temperature effects may originate from the modulation of TRPV1 on vascular smooth muscle cells (vSMCs), where channel activation promotes arteriole constriction. Here, we ask which of these pathways is most responsible for the body temperature perturbations elicited by TRPV1 drugs in vivo. We address this question by selectively eliminating TRPV1 expression in sensory neurons or vSMCs and show that only the former abrogates agonist-induced hypothermia and antagonist-induced hyperthermia. Furthermore, lesioning the central projections of TRPV1-positive sensory nerve fibers also abrogates drug-mediated thermomodulation, whereas eliminating CGRP has no effect. Thus, TRPV1 drugs alter core body temperature by modulating sensory input to the central nervous system, rather than through peripheral actions on the vasculature. These findings suggest how mechanistically distinct TRPV1 antagonists may diminish inflammatory pain without affecting core body temperature.

**\*For correspondence:**
David.Julius@ucsf.edu

## Editor's evaluation

This study establishes that the activity of TRPV1 channels in nociceptive neurons and their synaptic inputs to the central nervous system are required for the effects on body temperature of TRPV1-acting agonists and antagonists, whereas the activity of TRPV1 channels in the vasculature is dispensable, as is the release of neuropeptides from the peripheral nociceptive terminals. This conclusion is achieved by combining genetic tools to selectively ablate channel expression from either nociceptors or the vasculature, behavioral assays, pharmacology, and calcium imaging. The results have relevant implications for the physiology of pain, thermoregulation, and inflammatory processes, as well as for the clinical development of TRPV1-targeted analgesics.

## Introduction

The detection of painful stimuli begins with the activation of peripheral sensory neurons known as nociceptors. The capsaicin (vanilloid) receptor, TRPV1, is a nonselective cation channel expressed by a subset of small diameter unmyelinated (C) and medium diameter myelinated (Aδ) nociceptors (*Basbaum et al., 2009*). TRPV1 is activated by heat (*Cao et al., 2013*; *Caterina et al., 1997*) and is

modulated by inflammatory mediators, including extracellular protons (*Jordt et al., 2000*; *Tominaga et al., 1998*) and bioactive lipids (*Cao et al., 2013*), making it one of the key polymodal receptors for detecting noxious stimuli. Many studies have found that TRPV1 contributes not only to acute pain, but also to persistent pain conditions, particularly those associated with inflammation (*Julius, 2013*). As such, TRPV1 has become an actively pursued drug target for analgesic therapy. Unfortunately, this pursuit has been stymied by the fact that many TRPV1 modulators produce core body temperature changes.

Injection or consumption of capsaicin, the active component of chili peppers and a specific TRPV1 agonist, drives a range of thermoregulatory processes, including vasodilation, sweating, and panting, all of which promote a decrease in core body temperature (*Jancsó-Gábor et al., 1970a*; *Szolcsányi, 2015*). Conversely, many TRPV1 antagonists, including drug candidates, induce hyperthermia both in animal models and in human clinical trials. For example, oral administration of AMG517 increases body temperature of some participants to 39–40°C for 1–4 days (*Gavva et al., 2008*). Other TRPV1 antagonists, for example, ABT-102 (*Othman et al., 2013*; *Rowbotham et al., 2011*), AZD1386 (*Gomtsyan and Brederson, 2015*; *Quiding et al., 2013*), and JNJ-39439335 (aka. Mavatrep) (*Manitpisitkul et al., 2016*), also produce varying degree of hyperthermia at therapeutic dosages.

It is unclear what site and mechanism of action underlie these thermoregulatory effects of TRPV1-selective drugs. While nociceptors are the main site of TRPV1 expression, functional channels are also found on vascular smooth muscle cells (vSMCs) within a subset of terminal arterioles in peripheral (but not central) vasculature (*Cavanaugh et al., 2011*; *Phan et al., 2020*). The activation of vascular TRPV1 leads to vasoconstriction (*Cavanaugh et al., 2011*; *Phan et al., 2022*; *Phan et al., 2020*), which is generally thought to be associated with decreased blood flow and enhanced heat conservation. On the other hand, vasoconstriction at sites outside of thermoregulatory tissues may shunt blood flow into these areas and promote heat dissipation. Accordingly, it is unknown whether and how vascular TRPV1 participates in drug-mediated body temperature modulation, particularly in comparison with neuronal TRPV1. Also at issue is the relative contribution of the central versus peripheral consequences of the activation of primary afferents. Sensory neuron activation of central pathways involves spinal cord and trigeminal circuits that project to thermoregulatory centers in the brain, which can then mobilize adaptive physiological or behavioral programs, including the regulation of sympathetic outflow (*Alawi et al., 2015*) for control of thermogenesis, vascular tone, etc. Once activated, subsets of primary sensory neurons also release transmitters peripherally. The neuropeptide calcitonin gene-related peptide (CGRP) is most relevant here as it elicits pronounced local vasodilation following activation of TRPV1 on perivascular sensory nerves (*Argunhan and Brain, 2022*; *Zygmunt et al., 1999*).

Here, we use genetic and pharmacological approaches to assess the relative contribution of TRPV1 on arterioles versus sensory neurons to agonist- or antagonist-evoked changes in core body temperature. We find that TRPV1 on sensory neurons predominates in mediating these actions, and furthermore, that this requires signal transmission to the central nervous system, but not peripheral release of CGRP.

## Results

### Neuronal TRPV1 is essential for drug-mediated changes in core body temperature

To determine whether drug-evoked body temperature change is mediated by TRPV1 in peripheral nociceptors versus vSMCs, we generated mice lacking TRPV1 in each of these locales. First, *Trpv1*[fl/fl] mice were produced by inserting a pair of loxP sites flanking the second exon of the *Trpv1* gene, which contains the initiator ATG codon (*Figure 1A*). Crossing this line with *Pirt*[Cre] (*Anderson et al., 2018*) or *Tg(Myh11-CreERT2)* (*Wirth et al., 2008*) mice allowed for specific elimination of TRPV1 expression in primary sensory neurons or vSMCs, respectively. We confirmed the selectivity of these genetic manipulations by immunohistochemical staining (*Figure 1—figure supplement 1*) as well as by functional analysis using calcium imaging, thereby demonstrating loss of TRPV1 expression and/or capsaicin sensitivity within dorsal root ganglion (DRG) neurons or ear skin arteriole segments harvested from these animals (*Figure 1B, C*). Both cell type-specific knockout lines exhibited normal basal core body temperature and showed no noticeable peculiarities in the circadian pattern of body temperature (*Figure 1—figure supplement 2*). These phenotypes resemble those observed in global

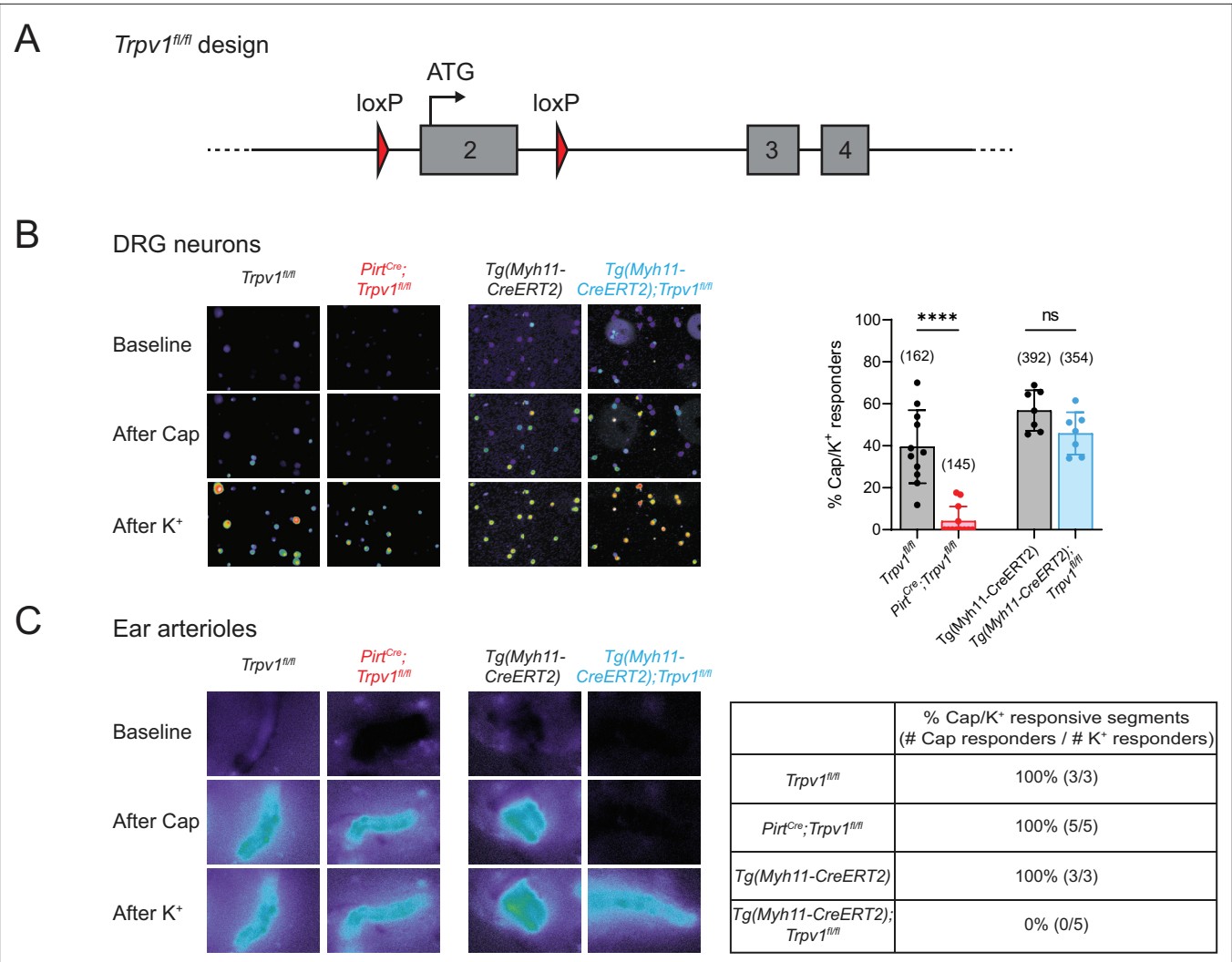

**Figure 1.** Tissue-specific knockout selectively abolishes functional expression of TRPV1 in sensory neurons or vascular smooth muscle cells. (**A**) Design of the Trpv1-floxed mouse line. Numbered gray boxes indicate exons. (**B**) Calcium responses of Fura2-AM-loaded dorsal root ganglion (DRG) neurons from sensory neuron-specific knockouts (*Pirt^{Cre};Trpv1^{fl/fl}*), smooth muscle-specific knockouts (*Tg(Myh11-CreERT2);Trpv1^{fl/fl}*), and their respective genotype controls following capsaicin (5 μM) application. In these and all following experiments, control animals received same tamoxifen treatment as knockouts. High (~50 mM) extracellular $K^+$ was used to reveal all neurons in the imaging field. Left panels show representative images of 340/380 emission ratio in rainbow scale; summary data are in bar graph on right. Each datapoint represents one experimental preparation. Total numbers of $K^+$ responders are indicated in brackets. Statistical analyses by unpaired, two-tailed Welch's *t*-test; ****: $p \leq 0.0001$, ns: not significant. (**C**) Calcium responses of Fura2-AM-loaded ear arterioles of the indicated genotypes to capsaicin (10 μM) and high extracellular $K^+$ application. Table on right shows the percentage of capsaicin-responsive segments among $K^+$-responsive arteriole segments, with respective number of segments given in brackets. Source data for (**B**) can be found in *Figure 1—source data 1*.

The online version of this article includes the following source data and figure supplement(s) for figure 1:

**Source data 1.** Calcium imaging of DRG neurons from tissue-specific Trpv1 knockouts and controls (related to *Figure 1B*).

**Figure supplement 1.** Representative images of dorsal root ganglion (DRG) sections from mice of indicated genotypes immunostained with anti-TRPV1 antibody (green).

**Figure supplement 2.** Average core body temperature (**A**) and circadian body temperature fluctuations (**B**) of the indicated genotypes.

**Figure supplement 2—source data 1.** Average core body temperature of tissue-specific Trpv1 knockouts and controls (related to *Figure 1—figure supplement 2F*).

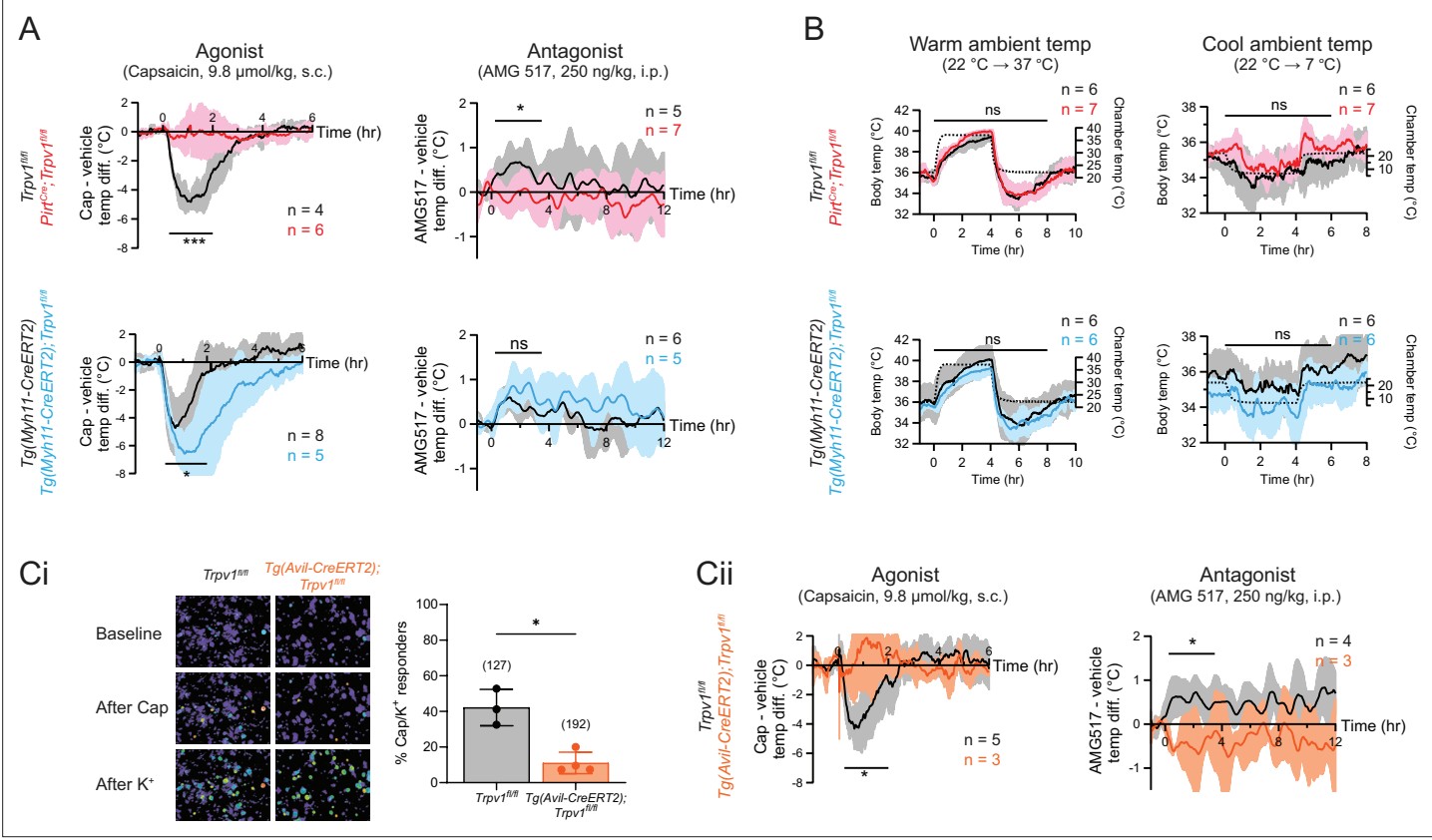

**Figure 2.** Neuronal, rather than vascular, TRPV1 mediates drug-evoked thermomodulatory effects. (**A**) Telemetric recordings of core body temperatures of $Pirt^{Cre};Trpv1^{fl/fl}$ mice (red), $Tg(Myh11\text{-}CreERT2);Trpv1^{fl/fl}$ mice (blue), and their respective genotype controls (black). Traces show responses to subcutaneous injection of capsaicin (9.8 μmol/kg) and intraperitoneal injection of AMG517 (250 nmol/kg) administered at time 0, after subtracting measurements from vehicle-alone trials. Baselines (mean temperature across the 1-hr window before drug administration) have been offset to zero for comparison. Responses to AMG517 were smoothened by taking running averages to better display trends. All traces are mean ± standard deviation (SD). Black bars above traces indicate the time windows over which statistical comparisons (mixed effects analyses) between genotypes were made. (**B**) Body temperature response of the above genotypes to changes in ambient temperature (dotted line, right axis). Statistical comparisons by mixed effects analyses. Source data are in *Figure 2—source data 2*. (**Ci**) Representative images (left) and quantification (right) of dorsal root ganglion (DRG) neurons from $Tg(Avil\text{-}CreERT2);Trpv1^{fl/fl}$ and control mice that are responsive to capsaicin (5 μM) and high extracellular K[+]. Source data are in *Figure 2— source data 3*. (**Cii**) Body temperature recordings as in (**A**) but from $Tg(Avil\text{-}CreERT2);Trpv1^{fl/fl}$ mice (orange) and controls (black). Source data for (**A**) and (**Cii**) can be found in *Figure 2—source data 1*. For all statistical tests, *: p ≤ 0.05; ***: p ≤ 0.001; ns: not significant.

The online version of this article includes the following source data and figure supplement(s) for figure 2:

**Source data 1.** Core body temperature responses of tissue-specific Trpv1 knockouts and controls to Trpv1 drugs (related to *Figure 2A and Cii*).

**Source data 2.** Core body temperature responses of tissue-specific Trpv1 knockouts and controls to ambient temperature changes (related to *Figure 2B*).

**Source data 3.** Calcium imaging of DRG neurons from inducible, sensory neuron-specific Trpv1 knockouts and controls (related to *Figure 2Ci*).

**Figure supplement 1.** Core body temperature comparisons between drug- and vehicle-injected trials.

**Figure supplement 1—source data 1.** Raw core body temperature data for comparisons between drug- and vehicle-injected trials (related to *Figure 2—figure supplement 1*).

TRPV1 knockout mice (*Garami et al., 2011*; *Iida et al., 2005*; *Szelényi et al., 2004*), likely reflecting redundant or compensatory actions of other thermosensitive receptors in these animals (*Solinski and Hoon, 2019*; *Tan and Knight, 2018*).

To examine the acute effects of TRPV1 modulation in wild type or mutant mice, we administered an agonist (capsaicin, 9.8 μmol/kg, single dose, s.c.) and recorded core body temperature changes with telemetric probes implanted in the abdomen. Consistent with previous observations (*Caterina et al., 2000*), capsaicin-induced robust hypothermia of ~5°C at peak (~1 hr postinjection) in control (*Trpv1^{fl/fl}* or *Tg(Myh11-CreERT2)*) mice. Strikingly, this capsaicin-evoked hypothermic response was

completely abolished in sensory neuron-specific knockouts (*Pirt^Cre;Trpv1^fl/fl*) (see **Figure 2A**, upper left for genotype comparison and **Figure 2—figure supplement 1** for vehicle vs. drug comparison). These results recapitulate the phenotype observed in global TRPV1 knockouts (**Caterina et al., 2000**). Unexpectedly, the smooth muscle-specific knockout (*Tg(Myh11-CreERT2);Trpv1^fl/fl*) (**Figure 2A**, lower left) showed a robust hypothermic response that was even more pronounced in both amplitude and duration compared to controls. Together, these observations show that agonist-evoked hypothermia requires TRPV1 expressed by sensory nerve fibers, but the activation of channels on the vasculature has an opposing effect, likely related to vasoconstriction that limits radiant heat loss.

We next recorded core body temperature responses following administration of AMG517 (250 ng/ kg, 3 doses, i.p.), which we selected as a representative TRPV1 antagonist because its hyperthermic effect has been verified to be on-target (i.e., absent in *Trpv1^−/−* mice) and its dose–response relationship has been documented previously (**Garami et al., 2010**). Indeed, AMG517 elicited a characteristic hyperthermia in control animals (as in human clinical trials), reaching a peak of ~0.5°C within 2 hr after injection (cf. **Garami et al., 2010**). This response was abolished in *Pirt^Cre;Trpv1^fl/fl* mice but did not change significantly in *Tg(Myh11-CreERT2);Trpv1^fl/fl* animals (**Figure 2A**, right column and **Figure 2— figure supplement 1**). Loss of TRPV1 in either cell type did not impact body temperature regulation in response to changes in ambient temperature (**Figure 2B**), suggesting that relevant thermosensory mechanisms (e.g., contribution from other thermosensitive TRP channels) and thermoeffector pathways (**Tan and Knight, 2018**) function normally in these animals and are unlikely to explain the differential pharmacological responses that we observed across genotypes.

To rule out developmental factors that could account for the observed lack of capsaicin/AMG517-evoked responses in our neuronal TRPV1 knockout, we also studied an *Tg(Avil-CreERT2)* (**Lau et al., 2011**) line to ablate TRPV1 in primary sensory neurons after 8 weeks of age. Functional ablation was less effective and more variable in *Tg(Avil-CreERT2);Trpv1^fl/fl* mice compared to *Pirt^Cre;Trpv1^fl/fl* animals (based on calcium imaging, 11.1 ± 6.0% of DRG neurons remained capsaicin responsive after tamoxifen treatment compared to 42.2 ± 10.2% in *Trpv1^fl/fl* controls) (**Figure 2Ci** and **Figure 1—figure supplement 1**). Nonetheless, capsaicin/AMG517-induced body temperature changes were similarly abrogated in these inducible knockouts (**Figure 2Cii**). These data further support the conclusion that sensory TRPV1 predominates in mediating drug-evoked thermomodulatory responses.

## Body temperature perturbations require central transmission, but not peripheral CGRP release

When activated, primary afferent nociceptors release transmitters from their central terminals to convey signals to the spinal cord and brain. However, these neurons are unique in that they also release transmitters peripherally to initiate neurogenic inflammatory processes, including CGRP-evoked vasodilation. Importantly, such peripheral release occurs independently of central transmission.

We therefore asked whether peripheral or central transmission from TRPV1⁺ nociceptors is required for the capsaicin/AMG517-evoked responses. To address this question, we selectively ablated the central projections of TRPV1⁺ DRG neurons by intrathecal (IT) injection of capsaicin (10 μg/mouse; control mice received vehicle). Successful ablation of central TRPV1⁺ terminals was validated by decreased nocifensive behavior on a hotplate (**Figure 3A**), as previously described (**Cavanaugh et al., 2009**). This procedure leaves peripheral projections of TRPV1⁺ fibers intact and capable of releasing proinflammatory agents (**Lin King et al., 2019**). Indeed, intraplantar injection of the TRPA1 agonist, allyl isothiocyanate (AITC), which activates a subset of TRPV1⁺ nociceptors (**Jordt et al., 2004**), failed to induce nocifensive behaviors in animals treated intrathecally with capsaicin (**Figure 3A**). However, AITC elicited robust local edema in both Cap-IT and Veh-IT control mice (**Figure 3A**). Interestingly, capsaicin/AMG517-evoked body temperature responses were significantly reduced in mice with ablated central projections (**Figure 3B**), indicating that central transmission from TRPV1⁺ DRG fibers is required for drug actions. A small residual effect of these drugs was observed (**Figure 3B**), which could reflect contributions from incompletely ablated central projections, vagal or trigeminal afferents, or sustained peripheral release of the vasodilative neuropeptide, CGRP. To address this last possibility, we examined mice lacking CGRP (i.e., *Calca^−/−* generated by crossing *Calca^CreERT2* to homozygosity) (**Song et al., 2012**; **Figure 3—figure supplement 1**) and found that they showed normal body temperature responses to capsaicin or AMG517 (**Figure 3C**), suggesting that peripheral CGRP release from TRPV1⁺ nociceptors is largely

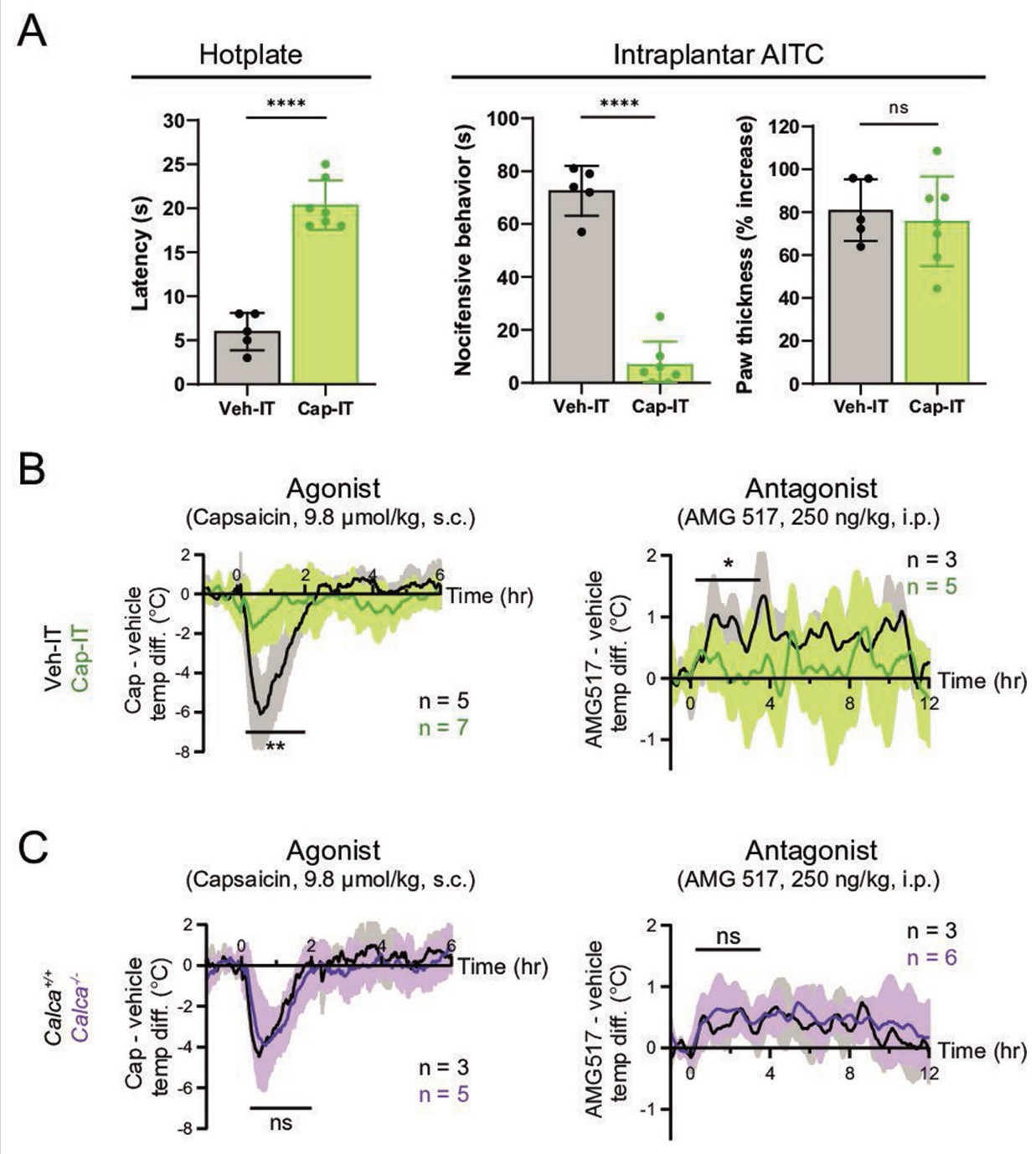

**Figure 3.** Thermomodulatory effects of TRPV1-selective drugs require central transmission, but not peripheral calcitonin gene-related peptide (CGRP) release. (**A**) Comparison of nocifensive behaviors of mice injected intrathecally with capsaicin (Cap-IT, green, 10 μg/mouse, *n* = 7) versus vehicle controls (black, *n* = 5). Cap-IT mice were significantly slower (i.e., had higher latency) in showing the first nocifensive response in a hotplate test (left). Cap-IT mice also spent significantly less time in nocifensive behaviors (middle) after intraplantar AITC injection despite similar levels of local edema (right), suggesting successful selective ablation of TRPV1[+] nociceptors' central projections. Unpaired, two-tailed Welch's *t*-test in all panels. See *Figure 3—source data 1* for source data. (**B**) Core body temperature recordings showing diminished hypo- and hyperthermic responses to capsaicin and AMG517, respectively, in Cap-IT mice. (**C**) Normal hypo- and hyperthermic responses to capsaicin and AMG517, respectively, in *Calca*[−/−] mice. All traces are plotted and statistical tests carried out as in *Figure 2*. For all statistical tests, *: p ≤ 0.05; ****: p ≤ 0.0001; ns: not significant. Source data for (**B**) and (**C**) are in *Figure 3—source data 2*.

The online version of this article includes the following source data and figure supplement(s) for figure 3:

**Source data 1.** Nocifensive behaviors of Cap-IT and Veh-IT mice (related to *Figure 3A*).

*Figure 3 continued on next page*

*Figure 3 continued*

**Source data 2.** Core body temperature responses to Trpv1 drugs of Cap-IT and Calca-/- mice as well as controls (related to *Figure 3B and C*).

**Figure supplement 1.** Successful ablation of calcitonin gene-related peptide (CGRP) expression in *Calca⁻/⁻* mouse line as shown by the lack of anti-CGRP immunosignal (green) in an example dorsal root ganglion (DRG) section from a *Calca⁻/⁻* mouse compared to a wild-type littermate control.

dispensable in this drug-mediated thermomodulatory process, consistent with a centrally mediated mechanism.

## Discussion

In the past two decades, TRPV1 antagonists have been actively pursued by the pharmaceutical industry to treat diverse persistent pain conditions. Despite analgesic efficacy in some indications, such as osteoarthritic knee pain, two main 'on-target' adverse side effects have limited further development of these drugs for clinical use: diminished sensitivity to potentially injurious heat, and increased core body temperature (*Bamps et al., 2021*; *Garami et al., 2020*; *Gomtsyan and Brederson, 2015*). Our current work addresses the mechanism underlying the latter adverse effect. Our findings show that the thermomodulatory responses of mice to acute systemic activation/inhibition of TRPV1 are mediated predominantly by the central projections of TRPV1-expressing nociceptors and thus likely involve output from thermoregulatory centers in the hypothalamus, which may recruit the sympathetic system, for example, with the engagement of adrenergic receptors (*Alawi et al., 2015*). On the other hand, the concurrent and robust agonist-evoked hypothermia in smooth muscle-specific knockouts reveals an opposing action by vascular TRPV1 that may help constrain hypothermic responses within a tolerable temperature or temporal window. Incidentally, *Sharif-Naeini et al., 2008* have implicated supraoptic hypothalamic expression of an N-terminal variant of TRPV1 in anticipatory vasopressin release during hyperthermia. Moreover, central injection of TRPV1 agonists have also been reported to trigger transient hypothermia (*Hori, 1984*; *Jancsó-Gábor et al., 1970b*; *Muzzi et al., 2012*) whereas intracerebroventricular injection of a TRPV1 antagonist prolonged hyperthermia induced by exposure to a 35°C environment (*Yonghak et al., 2020*). However, the precise site of action of such pharmacological manipulations has not been unambiguously demonstrated. Most importantly, our finding that ablation of the central projections of sensory neurons prevents the thermomodulatory side effects of pharmacological manipulation of TRPV1 argues that any hypothalamic expression is not sufficient to sustain these effects in vivo.

In addition, our results suggest a rationale for developing TRPV1 antagonists that would selectively or preferentially inhibit peripheral release of proinflammatory transmitters, without compromising the central pathway and thus thermoregulation. This idea is related to our recent analysis of a scorpion toxin that stabilizes the TRPA1 channel in a unique open state with a low $Ca^{2+}$ permeability (*Lin King et al., 2019*). Thus, the application of the toxin to nociceptors in vivo promotes neuronal transmission, but not the $Ca^{2+}$-dependent release of CGRP, hence eliciting pain without neurogenic inflammation. Similarly, mode-specific modulation of TRPV1 could, in principle, produce analgesia by preferentially inhibiting channel states with high $Ca^{2+}$ permeability, thereby reducing neuropeptide release from peripheral terminals to diminish neurogenic inflammation. Interestingly, the $Ca^{2+}$ fraction of TRPV1 current is apparently smaller when the channel is activated by low pH, rather than by capsaicin (*Samways et al., 2008*), supporting the existence of mode-specific states. Moreover, recent generation of TRPV1 antagonists that do not block proton-evoked channel activation appear less likely to alter core body temperature (*Garami et al., 2010*) (see *Bamps et al., 2021* also for review). In fact, one such compound, NEO6860, has recently been tested on patients with osteoarthritis knee pain and found not to be hyperthermic, although its analgesic effect was marginal in this one trial (*Arsenault et al., 2018*). Future clinical studies will determine whether such mode-specific drugs are efficacious in treating inflammatory pain while sparing acute heat sensitivity and thermoneutrality.

## Materials and methods
### Animals

All animal experiments were conducted according to protocols (AN192533 and AN183265) approved by the Institutional Animal Care and Use Committee at UCSF. *Pirt^Cre* mice were kindly provided by Dr.

Xinzhong Dong at Johns Hopkins University. *Tg(Myh11-CreERT2)*, *Tg(Avil-CreERT2)*, and *Calca^CreERT2* mice were obtained from the Jackson Laboratory. *Calca^CreERT2* mice, in which the ATG translational initiator of the *Calca* gene is replaced by a CreERT2 sequence, were crossed to homozygosity to produce *Calca^{-/-}*. For *Tg(Myh11-CreERT2)* induction, regular food was replaced with chow containing 400 mg/kg tamoxifen citrate (Harlan Teklad, TAM400) for 1 week when mice were at 5 weeks of age. For *Tg(Avil-CreERT2)* induction, mice were injected with 2 mg tamoxifen/day (Sigma-Aldrich, T5648, 20 mg/ml in corn oil) for 5 consecutive days at 8 weeks of age and 5 mg tamoxifen/day for another 5 consecutive days at 10 weeks of age to ensure effective knockout. Mice were raised under 12:12 light–dark cycles with ad libitum access to food and water. In all experiments, we used male mice that were 2–5 months old and randomly selected. Males were chosen because the *Myh11-CreERT2* allele is integrated into the Y chromosome. Age-matched animals (littermates in most cases) were used as genotype controls and were subjected to the same tamoxifen treatment. No notable differences in body size were observed across genotypes.

## Generation of *Trpv1^{fl/fl}* mice

Two loxP sites were inserted by the CRISPR/Cas system across the second exon of the *Trpv1* gene, where the ATG codon is located. Briefly, two crRNAs (5′-UCA AGG UGU CCU GAU UAA CGG UUU UAG AGC UAU GCU-3′ and 5′-CCA CAC UCU UGC GAU CUU GCG UUU UAG AGC UAU GCU-3′) were separately annealed to tracrRNA (IDT, #1072532) at 1:1 molar ratio in a thermocycler at 95°C for 5 min and then ramped down to 25°C at 5°C/min. The annealed oligos were mixed and incubated with Cas9 nuclease (IDT, #1081058) at a final concentration of 20 ng/µl (for both oligos and nuclease) at room temperature for 10–15 min to allow the formation of ribonucleoprotein (RNP) complexes. A synthesized DNA gene fragment spanning a homology region from 500 bp upstream to 500 bp downstream of the two loxP insert sites was cloned into pBluescript II KS(+) vector (Genscript). The gene fragment was released by EcoRI and KpnI digestion and was added as a repair template to the RNP complexes at a final concentration of 10 ng/µl. The mixture was filtered through a 0.1-µm pore centrifugal filter (Millipore, UFC30VV25) to remove any solid particles and was microinjected into mouse embryos. Animals carrying the targeted loxP insertions were identified by PCR on tail DNA and confirmed by sequencing. A set of three primers were used to genotype the *Trpv1* allele: Trpv1 forward, 5′-GTG TCA GCT CCC TCT CAA GG-3′; Trpv1WT reverse, 5′-GCC AGA CCA CCT CTG AAG GCT T-3′; Trpv1FL reverse, 5′-GCC AGA CCA CCT CTG AAA TAA C-3′. The primer pair of Trpv1 forward and Trpv1WT reverse gave a 1185-bp band for the WT allele, whereas the pair of Trpv1 forward and Trpv1FL reverse gave a 1253 bp band for the floxed allele. The *Trpv1^{fl/fl}* mouse line is available on reasonable request to the corresponding author.

## Calcium imaging

Mice were euthanized by $CO_2$ asphyxiation followed by decapitation according to American Veterinary Medical Association's guidelines. For calcium imaging of sensory neurons, DRGs were harvested from a mouse into ice-cold L-15 medium (Thermo Fisher Scientific, 11415064). The tissues were digested with papain (Worthington Biochemical, LS003126, 60 unit) for 10 min, followed by collagenase (Sigma-Aldrich, C9407, 12 mg/ml) and dispase I (Sigma-Aldrich, D4818, 15 mg/ml) for another 10 min, both at 37°C. The digested ganglia were then mechanically triturated in L-15 medium containing 10% fetal bovine serum to dissociate the neurons and were plated onto coverslips precoated with poly-D-lysine (Sigma-Aldrich, A-003-E) and mouse laminin (Thermo Fisher Scientific, 23017015). Neurons were cultured in Dulbecco's modified Eagle's medium with 4.5 g/l glucose (UCSF Media Production, DME-H21) and 10% horse serum overnight at 37°C under 5% $CO_2$. When used for calcium imaging, neurons were washed with Ringer's solution (140 mM NaCl, 5 mM KCl, 2 mM $MgCl_2$, 2 mM $CaCl_2$, 10 mM HEPES (4-(2-hydroxyethyl)-1-piperazineethanesulfonic acid), and 10 mM glucose, pH 7.4 with NaOH; 290–300 mOsm $kg^{-1}$) and loaded with 10 µg/ml Fura2-AM (Thermo Fisher Scientific, F1201, stock: 1 mg/ml in DMSO) in Ringer's solution containing 0.02% Pluronic F-127 (Thermo Fisher Scientific, P3000MP) at room temperature in the dark for 1 hr. After Ringer's washes, neurons were bathed in Ringer's solution and imaged under 340 and 380 nm excitation (Sutter, Lambda LS illuminator) at a time interval of 2 s with a FLIR Grasshopper3 camera. Images were acquired and analyzed with the MetaFluor software (Molecular Devices) in a blinded manner. Capsaicin (diluted to 10 µM from a 10 mM stock in DMSO) was delivered by manual pipetting to achieve a final concentration of 5 µM in

the bath. An equal volume of high-K⁺ Ringer's solution (same as Ringer's except for 5 mM NaCl and 140 mM KCl) was added to reveal neuronal identities. Multiple coverslips from at least two animals were examined for each genotype.

For calcium imaging of arterioles, third-order arterioles were isolated from ears in the following solution: 137 mM NaCl, 5.6 mM KCl, 1 mM $MgCl_2$, 0.42 mM $Na_2HPO_4$, 0.44 mM $NaH_2PO_4$, 4.2 mM $NaHCO_3$, 10 mM HEPES, and 1 mg/ml bovine serum albumin, pH 7.4 with NaOH (*Cavanaugh et al., 2011*). Arterioles were cut in pieces and laid on Cell-Tak-coated coverslips. Fura loading and calcium imaging were done in Ringer's solution as above, except that a 20 µM stock of capsaicin was used to achieve a final concentration of 10 µM in the bath.

## Telemetry

Mice were anesthetized and implanted each with a wireless temperature probe (Starr Life Sciences, G2 E-Mitter) in its abdominal cavity. Mice were allowed to recover for at least 1 week before their home cages were transferred onto Energizers/Receivers (Starr Life Sciences, ER4000) in a rodent incubator (Power Scientific, RIS33SD or RIS70SD) maintained at 22°C. Core body temperature was recorded with the VitalView software at a sampling rate of once per min. Capsaicin/AMG517 injections and chamber temperature changes began after at least 1 day of habituation in the incubator.

## Injections

For each mouse, a single dose of 9.8 µmol/kg capsaicin (Tocris, #0462, stock: 0.98 µmol/ml in 0.72% saline with 10% Tween 80 and 10% ethanol) was administered by subcutaneous injection. Vehicle was injected to the same animal 3 days before/after capsaicin injection. Since the hyperthermic effect of AMG517 is mild, in order to reveal the response amid random body temperature fluctuations, mice were subjected to three alternating sets of intraperitoneal injections of 250 ng/kg of AMG517 (Cayman Chemical Company, #26191, stock: 50 ng/ml in 100% ethanol) and vehicle, with any two injections spaced 3 days apart. Averages across the three trials were used. To avoid any circadian effect, all capsaicin and AMG517 injections were done at 10 AM. Some mice experienced both capsaicin and AMG517 challenges; no obvious difference in TRPV1 drug-evoked responses was seen in animals receiving one type versus two types of challenges, so data were pooled for analyses. To ablate the central projections of TRPV1⁺ sensory neurons, mice were anesthetized and intrathecally injected with 10 µg capsaicin (stock: 1 mg/ml in 0.72% saline with 10% Tween 80 and 10% ethanol). Cap-IT mice were allowed to recover for at least 1 week before telemetric recording.

## Nocifensive behavioral tests

Two sets of behavioral tests were used to confirm successful ablation of the central projections of sensory neurons. In the first, we measured the time latency to first nocifensive response (e.g., licking or flicking paws, jumping) when a mouse was placed on a 55°C hotplate. Mice were removed from the hotplate once nocifensive behaviors were observed or at a cutoff of 30 s, whichever was the shortest, to minimize tissue damage. Two sets of measurements were taken for each mouse on different days. In the second test, we injected 0.75% (vol/vol) allyl isothiocyanate (AITC; Sigma-Aldrich, #377430) in mineral oil intraplantar to the hindpaw after light isoflurane anesthesia. After recovery, we videorecorded the individually housed mice in transparent cylinders for 20 min and quantified the amount of time in which nocifensive behaviors were exhibited. Paw thickness was measured by digital calipers before and after AITC injection to assess the degree of local edema. Only mice that has passed the cutoff in at least one of the two hotplate trials and showed at least 50 s of nocifensive behaviors after AITC injection were used for telemetric measurements. The time latency for the hotplate experiment reported in *Figure 3A* is the average of the two measurements.

## Data analysis and statistics

Unless otherwise specified, collective data herein are reported as mean ± standard deviation (SD). Statistical analyses were done with the Prism software (GraphPad) and data are reported following the ARRIVE guidelines. For nocifensive behavioral tests, sample sizes were estimated by using power analyses based on variances obtained from previous experiments. In calcium imaging assays and nocifensive behavioral tests, comparisons across genotypes were made by using unpaired, two-tailed Welch's *t*-test, assuming that parametric approach is applicable based on previous similar experiments.

For telemetric measurements, mixed effects analyses were performed without assuming sphericity over the duration of body temperature change (indicated in plots). Mice that showed considerable body temperature fluctuations at baseline were excluded from analyses. For responses to AMG517, measurements were averaged across the three injection trials before comparisons. Significance was determined by the computed p value across genotypes. For all tests, $\alpha$ = 0.05. *p $\leq$ 0.05; **p $\leq$ 0.01; ***p $\leq$ 0.001; ****p $\leq$ 0.0001; ns: not significant. For display in figures, circadian body temperature fluctuations and responses to AMG517 were smoothened by taking running averages over a 30-min window before computing the mean and SD. This processing was done only for improving visualization (dampening random, fast fluctuations) and did not dramatically affect the kinetics of the slow AMG517 effect; statistical comparisons were done on raw data, not the smoothened data. Readers are referred to *Figure 2—figure supplement 1* for raw data, where absolute core body temperatures are presented for comparisons between drug- and vehicle-injected trials. Area under curve (AUC) on the side of the mean response peak was extracted from the raw data traces by Prism over the time window of 0–4 hr, with baseline defined to be the 1 hr stretch before drug administration. Unpaired, one-way ANOVA with multiple comparisons was carried out using the AUC values and statistical significance for the pairwise comparisons is reported.

## Immunohistochemistry

DRGs were acutely harvested into ice-cold L-15 medium and transferred into 4% paraformaldehyde for overnight fixation. After washes with phosphate-buffered saline (PBS, Quality Biological, 119-069-491), the ganglia were allowed to settle in 30% sucrose (Sigma-Aldrich, S7903) at 4°C and were then cryopreserved in Tissue-Tek O.C.T. Compound (Sakura Finetek USA) for sectioning at 10 µm thickness. Sections were washed with PBST (i.e., PBS containing 0.1% Triton X-100 [Sigma-Aldrich, T8787]) and incubated with blocking buffer (PBST containing 10% normal goat serum from Thermo Fisher Scientific, 16-210-064) at room temperature for 1 hr. Subsequently, sections were incubated with rabbit anti-TRPV1 (Alomone Labs, ACC-030, 1:400) or rabbit polyclonal anti-CGRP (Abcam, ab36001, 1:400), together with guinea pig polyclonal anti-NeuN (Synaptic Systems, #266004, 1:400) in blocking buffer at 4 °C overnight. Following several washes, the sections were incubated with goat anti-rabbit IgG Alexa Fluor-488 (Thermo Fisher Scientific, A-11034, 1:500) and goat anti-guinea pig IgG Alexa Fluor-597 (Thermo Fisher Scientific, A-11076, 1:500) in blocking buffer at room temperature for 1 hr. Finally, the sections were washed and mounted with Fluoromount-G (SouthernBiotech, 0100-01). Z-stack images were taken with a Nikon CSU-W1 spinning disk confocal microscope (UCSF Center for Advanced Light Microscopy). Maximal intensity projections were generated by Fiji.

## Acknowledgements

We thank X Dong (Johns Hopkins) for providing the *Pirt*[Cre] mice as well as AM Bertholet, TA Wang and J Nikkanen (UCSF) for technical advice. We also thank all members of the Julius lab for discussion and RA Nicoll for critical reading of the manuscript. This work was supported by the Howard Hughes Medical Institute Hanna Gray Fellowship and the Croucher Fellowship for Postdoctoral Research (to WWSY), the Larry L Hillblom Foundation Fellowship (to LY), and NIH grants (R35 NS105038 to DJ and R35 NS097306 to AIB).

## Additional information

### Competing interests

Allan I Basbaum: Reviewing editor, eLife. The other authors declare that no competing interests exist.

### Funding

| Funder | Grant reference number | Author |
| --- | --- | --- |
| Howard Hughes Medical Institute | Hanna Gray Fellowship | Wendy Wing Sze Yue |

| Funder | Grant reference number | Author |
| --- | --- | --- |
| Croucher Foundation | Fellowship for Postdoctoral Research | Wendy Wing Sze Yue |
| Larry L. Hillblom Foundation | Larry L. Hillblom Foundation Fellowship | Lin Yuan |
| National Institute of Neurological Disorders and Stroke | R35 NS097306 | Allan I Basbaum |
| National Institute of Neurological Disorders and Stroke | R35 NS105038 | David Julius |

The funders had no role in study design, data collection, and interpretation, or the decision to submit the work for publication.

## Author contributions

Wendy Wing Sze Yue, Conceptualization, Data curation, Formal analysis, Validation, Investigation, Visualization, Methodology, Writing – original draft, Project administration, Writing – review and editing; Lin Yuan, Conceptualization, Data curation, Validation, Investigation, Methodology, Writing – review and editing; Joao M Braz, Investigation, Methodology, Writing – review and editing; Allan I Basbaum, Resources, Supervision, Funding acquisition, Writing – review and editing; David Julius, Conceptualization, Resources, Supervision, Funding acquisition, Writing – original draft, Project administration, Writing – review and editing

## Author ORCIDs

Wendy Wing Sze Yue http://orcid.org/0000-0002-7736-4262
David Julius http://orcid.org/0000-0002-6365-4867

## Ethics

This study was performed in strict accordance with the recommendations in the American Veterinary Medical Association's guidelines. All of the animals were handled according to approved Institutional Animal Care and Use Committee (IACUC) protocols (AN192533 and AN183265) of the University of California – San Francisco.

## Decision letter and Author response

Decision letter https://doi.org/10.7554/eLife.80139.sa1
Author response https://doi.org/10.7554/eLife.80139.sa2

---

# Additional files

## Supplementary files

• MDAR checklist

## Data availability

All data generated or analyzed during this study are included in the manuscript and supporting file; Source Data files have been provided for main text figures, Figure 1 – figure supplement 1 and Figure 2 – figure supplement 1.

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
