## [Editor Report]

This study establishes that the activity of TRPV1 channels in nociceptive neurons and their synaptic inputs to the central nervous system are required for the effects on body temperature of TRPV1-acting agonists and antagonists, whereas the activity of TRPV1 channels in the vasculature is dispensable, as is the release of neuropeptides from the peripheral nociceptive terminals. This conclusion is achieved by combining genetic tools to selectively ablate channel expression from either nociceptors or the vasculature, behavioral assays, pharmacology, and calcium imaging. The results have relevant implications for the physiology of pain, thermoregulation, and inflammatory processes, as well as for the clinical development of TRPV1-targeted analgesics.

---

## [Decision Letter]

**Decision letter after peer review:**

Thank you for submitting your article "TRPV1 drugs alter core body temperature via central projections of primary afferent sensory neurons" for consideration by *eLife*. Your article has been reviewed by 3 peer reviewers, including Andrés Jara-Oseguera as Reviewing Editor and Reviewer #1, and the evaluation has been overseen by Kenton Swartz as the Senior Editor. The following individuals involved in review of your submission have agreed to reveal their identity: Mark A. Hoon (Reviewer #2); Susan D Brain (Reviewer #3).

The reviewers have discussed their reviews with one another, and the Reviewing Editor has drafted this to help you prepare a revised submission. All reviewers agreed that the data in the manuscript is interesting, relevant, and clearly presented, and that most conclusions of the study appear to be well supported by the experimental work. However, reviewers also identified important points that need to be described better, including providing additional information that validates some of the tools that were used in the study. These are outlined below.

Essential revisions:

1) A more nuanced description should be provided for the mechanism by which vascular TRPV1 channels could give rise hypo- or hyperthermia after treatment with agonists and antagonists, respectively. This is one of the possibilities that the study was designed to test, but this possibility already appears unlikely given the known vascular effects of TRPV1 channel activation.

2) Although the in vitro assays on dissociated cells are reassuring, what additional steps were taken to validate the Trpv1 f/f allele leads to loss of Trpv1 expression? For instance, were antibody staining performed on tissue from the two conditional knockout animals to confirm loss of Trpv1 expression in the appropriate cells? This is important given that results for Advil-creER Trpv1f/f/ mice are mentioned in the paper (~25% reduction), but no results are presented. The later results should be added to the manuscript.

3) What were the baseline real temperatures in the different strains of mice? Were they similar? Did they breed/gain weight etc normally? What were their weights when used? There is quite a wide range of ages, were they age matched? These details would be useful as an appendix. Why were male mice chosen and were experiments carried out in a blinded manner?

4) Absolute body-temperature graphs with the data for individual animals treated with drugs or vehicle should also be included, at least as a supplementary figure. It would strengthen the manuscript if statistical tests were performed between vehicle-treated and drug-treated animals, rather than between WT and KO animals. This would reflect more accurately whether the different genetic manipulations reduce the effect of drugs on body temperature.

5) The apparent opposite effects of drugs in the body temperature of AviI-CreER-;Trpv1 fl/fl animals relative to WT should be discussed.

6) The evidence in the studies for vasodilators released from Trpv1-neurons, not causing body temperature change, is not strong. It is unclear why CGRP has been chosen for clarifying the mechanism, rather than the other major neuropeptides released, Substance P, or any other sensory-derived transmitter, some of which may also act indirectly. A rationale should be provided for focusing on CGRP specifically. The text should also be appropriately altered and/or wording toned down to acknowledge that the described studies with Calca knockout mice only provide evidence for this gene.

Additionally, the known peripheral microvascular vasodilator activity of CGRP is barely mentioned in the introduction, despite being one of the subjects under study and is not referenced in the results. This was discovered first in 1985.

7) Are there any thermoneutral TRPV1 antagonists available yet? There is some mention of this in the discussion, but please comment more thoroughly on this line of drug discovery research and the state of the art.

8) The role of the sympathetic system is barely referred to in the manuscript, but others consider the role of the sympathetic system is essential to the mechanism via which TRPV1 antagonists are hyperthermic. Additional discussion should be provided.

*Reviewer #1 (Recommendations for the authors):*

1) A more nuanced description of alternate hypothesis should be provided for how vascular TRPV1 channels could be expected to give rise hypo- or hyperthermia after treatment with agonists and antagonists. If this possibility is not physiologically feasible, given what is known about the expression and function of TRPV1 in the vasculature, this should be mentioned in the manuscript, which would need to be modified to reflect this point.

2) Absolute body-temperature graphs with the data for individual animals treated with drugs or vehicle should also be included, at least as a supplementary figure. It would strengthen the manuscript if statistical tests were performed between vehicle-treated and drug-treated animals, rather than between WT and KO animals. This would reflect more accurately the extent to which the specific deletion of TRPV1 channels decreases alterations in body temperature in response to drugs.

3) The apparent opposite effects of drugs in the body temperature of AviI-CreER-;Trpv1 fl/fl animals relative to WT should be discussed.

*Reviewer #2 (Recommendations for the authors):*

a. Although the in vitro assays on dissociated cells are reassuring, what additional steps were taken to validate the Trpv1 f/f allele leads to loss of Trpv1 expression? For instance, were antibody staining performed on tissue from the two conditional knockout animals to confirm loss of Trpv1 expression in the appropriate cells? This is important given that results for Advil-creER Trpv1f/f/ mice are mentioned in the paper (~25% reduction), but no results are presented. The later results should be added to the manuscript.

b. The evidence in the studies for vasodilators released from Trpv1-neurons, not causing body temperature change, is not strong. There are many vasodilators beyond the Calca gene which may be released by activated nociceptors including Trpv1-expressing neurons. For instance, inflammation might also arise from products of the Calcb and Tac1 genes, and other neuropeptides and other small molecules might be involved either directly or indirectly. Although this is not a central point of the study, it should be acknowledged that the described studies with Calca knockout only provide evidence for this gene. I.e., where these claims are made, the text should be appropriately altered to explain the limitations of this experiment and/or wording toned down.

*Reviewer #3 (Recommendations for the authors):*

1. Abstract.

Please include the nature of temperature change observed via activation of TRPV1.

Introduction.

'Hamstrung' - can an alternative word be used?

Are there any thermoneutral TRPV1 antagonists available yet, please can you comment on this line of drug discovery research and the state of the art. – There is some mention of this in the discussion.

I am unclear why CGRP has been chosen for clarifying the mechanism, rather than the other major neuropeptides released, Substance P (or any other sensory-derived transmitter). A better indication of the sensory transmitters released by TRPV1+ sensory nerves should be provided.

Additionally, the known peripheral microvascular vasodilator activity of CGRP is barely mentioned in the introduction, despite being one of the subjects under study and is not referenced in the results. This was discovered first in 1985.

Methods and Results.

The mice utilised appear to be well characterised, however data is lacking (Figure 1). I am impressed by the inclusion of the Avil-mice.

What were the baseline real temperatures in the different strains of mice? Were they similar? Did they breed/gain weight etc normally. What were their weights when used? I note quite a wide range of ages, were they age matched? These details, which I am sure the authors have, would be useful as an appendix. Why were male mice chosen and were experiments carried out in a blinded manner?

Discussion.

This is clear, however, the hypothalamus is concentrated on and evidence given. What about other evidence? For example concerning the brown adipose tissue and sympathetic system. Could this be put into context here? (e.g Alawi et al., 2015). This could allow an improved understanding of potential pathways.

---

## [Author Response]

Essential revisions:1) A more nuanced description should be provided for the mechanism by which vascular TRPV1 channels could give rise hypo- or hyperthermia after treatment with agonists and antagonists, respectively. This is one of the possibilities that the study was designed to test, but this possibility already appears unlikely given the known vascular effects of TRPV1 channel activation.

We thank the reviewer for requesting clarification of this issue, which we now more clearly address in the introduction. The main point is that TRPV1 is expressed only on a subset of arterioles, and while channel activation promotes vasoconstriction at these points, how this alters regional blood flow and perfusion of tissues is currently unknown. In other words, constricting the vasculature at these specific sites may increase or decrease blood flow to specific areas, including thermoregulatory zones such as the scalp, tail, ears, or other extremities. Thus, TRPV1 activation on arterioles could mediate hyper- or hypothermia, making vascular TRPV1 potentially relevant to the regulation of core body temperature.

2) Although the in vitro assays on dissociated cells are reassuring, what additional steps were taken to validate the Trpv1 f/f allele leads to loss of Trpv1 expression? For instance, were antibody staining performed on tissue from the two conditional knockout animals to confirm loss of Trpv1 expression in the appropriate cells? This is important given that results for Advil-creER Trpv1f/f/ mice are mentioned in the paper (~25% reduction), but no results are presented. The later results should be added to the manuscript.

We have included new data to validate the effects of our genetic manipulations on TRPV1 expression. As now shown in Figure 1 —figure supplement 1, we demonstrate lack of TRPV1 immunostaining in DRG sections from neuronal-specific (*Pirt-Cre* or *Avil-CreER*), but not arteriole-specific knockout mice (*Myh11-CreER*). We were unable to carry out a similar immunohistochemical analysis of vascular tissues due to low level TRPV1 expression relative to sensory ganglia, which in previously publications has been demonstrated using Trpv1-plap or Trpv1-Cre mouse lines that amplify the signal. Unfortunately, we cannot employ this strategy here because it is incompatible with our floxed allele.

In response to the comment concerning the reduction in capsaicin sensitivity in *Avil-CreER^+^;Trpv1^fl/fl^* mice, we show that the percentage of capsaicin-sensitive DRG neurons drops to 11.1% from the control value of 42.2% (i.e., a ~75% reduction). These data are shown in Figure 2Ci and we have now modified the text to make this clear. Importantly, these results are qualitatively consistent with our new histochemical data described above.

3) What were the baseline real temperatures in the different strains of mice? Were they similar? Did they breed/gain weight etc normally? What were their weights when used? There is quite a wide range of ages, were they age matched? These details would be useful as an appendix. Why were male mice chosen and were experiments carried out in a blinded manner?

Average baseline temperatures are presented in Figure 1 —figure supplement 2. No significant differences were found across genotypes. Moreover, no notable differences in body size were observed across genotypes.

We used age-matched animals in all our experiments, which in most cases were littermates.

The Myh11-CreER allele is integrated into the Y chromosome and so only males carry the transgene. For this reason, all comparisons were carried out with males.

These experimental details are now explicitly noted in the Materials and methods section.

4) Absolute body-temperature graphs with the data for individual animals treated with drugs or vehicle should also be included, at least as a supplementary figure. It would strengthen the manuscript if statistical tests were performed between vehicle-treated and drug-treated animals, rather than between WT and KO animals. This would reflect more accurately whether the different genetic manipulations reduce the effect of drugs on body temperature.

We thank the reviewers for requesting these data (now included as Figure 2 —figure supplement 1), which strengthen our conclusions.

5) The apparent opposite effects of drugs in the body temperature of AviI-CreER-;Trpv1 fl/fl animals relative to WT should be discussed.

This apparent drop below baseline in core body temperature following administration of AMG517 to *Avil-CreER^+^;Trpv1^fl/fl^* mice (Figure 2Cii) is not statistically significant, as shown in the new Figure 2 —figure supplement 1B.

6) The evidence in the studies for vasodilators released from Trpv1-neurons, not causing body temperature change, is not strong. It is unclear why CGRP has been chosen for clarifying the mechanism, rather than the other major neuropeptides released, Substance P, or any other sensory-derived transmitter, some of which may also act indirectly. A rationale should be provided for focusing on CGRP specifically. The text should also be appropriately altered and/or wording toned down to acknowledge that the described studies with Calca knockout mice only provide evidence for this gene.Additionally, the known peripheral microvascular vasodilator activity of CGRP is barely mentioned in the introduction, despite being one of the subjects under study and is not referenced in the results. This was discovered first in 1985.

As we now try to make clear, we have focused on CGRP because this is the main and the most potent vasodilator released from TRPV1-expressing peptidergic C-fibers. Moreover, we and others have shown that the vasodilatory actions of TRPV1 agonists are abrogated by CGRP receptor antagonists, making this agent especially relevant. But in any case, the most important result is that ablation of central projections of TRPV1-expressing neurons eliminates drug-evoked effects described here, ruling out a major contribution from peripherally released factors such as CGRP. Thus, we believe that these experiments, when taken together, support our main conclusions.

7) Are there any thermoneutral TRPV1 antagonists available yet? There is some mention of this in the discussion, but please comment more thoroughly on this line of drug discovery research and the state of the art.

A new compound, NEO6860, has been reported to be thermoneutral in an exploratory phase II clinical trial, but its analgesic efficacy on osteoarthritic knee pain has not reached significance compared to placebo. Further clinical studies have apparently been planned to test higher dosages. Interestingly, this new drug has a differential action on capsaicin versus proton-evoked channel activation, which is consistent with a hypothesis raised in our discussion. We now describe these observations in our revised manuscript and refer the reader to a recent review that discusses, in greater detail, progress in TRPV1 antagonist development. We thank the reviewers for prompting us to include this interesting point.

8) The role of the sympathetic system is barely referred to in the manuscript, but others consider the role of the sympathetic system is essential to the mechanism via which TRPV1 antagonists are hyperthermic. Additional discussion should be provided.

We certainly agree that regulation of sympathetic output in response to activation of TRPV1 afferents is a key component of the thermoregulatory response. We now mention this in both the introduction and discussion with appropriate references.